# Variability of a job search indicator induced by operationalization decisions when using digital traces from a meter

Carlos Ochoa📶*, Melanie Revilla📶

RECSM, Universitat Pompeu Fabra, Barcelona, Spain

\* carlos.ochoa@upf.edu

## Abstract

Digital traces —particularly metered data—offer researchers a valuable alternative to surveys for studying online behavior. However, because the concepts being measured are not directly observable in the data, their operationalization requires multiple decisions—for example, which events (e.g., visited websites) represent the concepts and which metrics (e.g., visit counts or time spent) capture their intensity. Using metered data from 600 Netquest panelists in Spain, this study investigates how operationalization choices affect the measurement of job search intensity. By varying metrics—combinations of measurement targets (such as sessions on job platforms or job offer pages) and measurement types (such as visit counts or time spent)—along with other factors (e.g., methods for separating search activity into spells or handling outliers), the study explores 10,080 operationalizations. Results reveal significant variability, with correlations between measurement pairs ranging from 0.14 to 0.91. Metrics sharing the same measurement target (e.g., sessions, job offer pages) demonstrate stronger convergent validity than those sharing only the same measurement type (e.g., time or visits). Other operationalization factors, such as session segmentation methods, also influence results, though less than metric choice. Importantly, operationalization decisions can affect substantive findings.

## 1. Introduction

Designing a survey requires numerous decisions that influence data quality by introducing measurement and representation errors [1]. In particular, many of the choices that affect measurement error stem from researchers' operationalization of concepts of interest —that is, the translation of concepts they intend to measure into survey questions [2]. For each concept, researchers must carefully: (1) define the concept and the specific dimension to be measured, (2) determine the wording of the question(s), (3) design appropriate response scale(s), and (4) decide whether to include contextual elements such as introductory text, definitions, instructions, or motivational

**Data availability statement:** The data and materials are available at: https://doi.org/10.17605/OSF.IO/SK4A5.

**Funding:** This project received funding from the European Research Council (ERC) under the European Union's Horizon 2020 research and innovation program (grant agreement No. 849165). The funders had no role in study design, data collection and analysis, decision to publish, or preparation of the manuscript.

**Competing interests:** The authors have declared that no competing interests exist.

prompts. Each step involves multiple choices that can introduce errors affecting data validity and reliability.

When using digital trace data instead of surveys, this process differs substantially. Digital trace data consist of records of activity generated through interactions with digital systems [3], including browsing history, search queries, social media interactions, GPS data, and app usage. These data eliminate memory-related errors and reduce participant burden, two common challenges in survey research. Among the various types of digital traces, metered data [4] are increasingly used to study online behaviors in fields such as media studies, political science, and social research [5]. Metered data consist of continuous records of internet activity, including at least URLs visited, collected through tracking software ("meters") installed voluntarily by participants on their browsing devices. Metered data are typically collected by opt-in online panels such as Netquest (www.netquest.com) and YouGov (www.yougov.com) or academic probabilistic panels like the GESIS panel [6]. Depending on the specific meter used, additional information can be obtained, including search queries, app usage, web and app page contents, and behavioral metrics such as time spent on online activities.

Due to their adaptability and the wide range of online behaviors they capture, metered data are among the most versatile forms of digital trace data for research [7]. However, despite some research treating metered data as a "source of truth" for studying online behaviors (e.g., [8–10]), they are not immune to errors. These errors stem from both technical issues (meter malfunctions, tracking pauses, system limitations) and user behaviours (device sharing, switching to non-metered devices). [11]. Some of these errors can be substantial. For instance, Bosch et al. [12], in a study using metered data in Spain, Portugal and Italy, found that more than 70% of participants were affected by tracking undercoverage, meaning that only part of their online activity was tracked.

Unlike survey data, digital trace data—including metered data—are collected passively, removing the need for survey question design [13]. However, operationalizing key concepts remains necessary, albeit in a different form. Instead of selecting question wording or response scales, researchers must determine, among other factors, which digital trace sources to use, which devices to track (e.g., PC or mobile), and which observed digital behaviors best measure the concept of interest.

This paper examines **how different approaches to operationalizing and measuring concepts using metered data can impact research outcomes**. The chosen substantive context for this analysis is online job search—an activity with significant implications for equal employment opportunities across demographic groups (e.g., gender, ethnicity) and for broader economic stability and growth [14,15]. Since the advent of the internet, the job-seeking landscape has been deeply reshaped [16], attracting researchers interested in measuring various aspects such as the effectiveness of online methods compared to traditional ones [17] or the usage of diverse online job search platforms [18]. However, assessing these aspects through surveys is challenging, as participants often struggle to accurately recall and thus report their behaviors [19]. For example, measuring the total time spent job searching online

over an extended period via a retrospective survey is prone to memory errors, given that job searching typically involves numerous short, repetitive interactions—such as browsing job platforms, reading postings, and deciding whether to apply. Alternatively, requiring participants to keep a daily diary to minimize memory errors imposes a high burden, potentially leading to non-participation or inaccurate reporting.

In such contexts, digital trace data offer a clear advantage over surveys by eliminating memory errors and reducing participant burden. Indeed, various forms of digital traces have been used to study job search behaviors, both as stand-alone measures (see "Background" section) and as a trigger for "in-the-moment" survey invitations [20]. In particular, Ochoa and Revilla [21], using metered data from 600 individuals in Spain over a nine-month period (March to November 2023), explored the potential of metered data for studying online job search behavior, operationalizing and measuring 12 job search-related concepts. These approaches have demonstrated that digital trace data can offer more accurate and granular insights compared to conventional survey-based methods, especially when tracking ongoing, repetitive activities like job searching.

Building on Ochoa and Revilla's [21] work and using the same dataset, this paper explores the different possible choices that could have been made to operationalize one of these concepts — job search intensity— and whether these choices affect the results. Job search intensity has long been conceptualized as the degree of effort and persistence individuals devote to finding employment. In classical labor economics, it reflects how much time/energy job seekers invest in search activities, influencing the likelihood and speed of re-employment [22,23]. Later research expanded this view, treating job search intensity as a multidimensional construct that encompasses various factors beyond time spent searching [24]. As job search activities increasingly take place online, the number of potentially relevant dimensions—observable through digital traces, such as the number of job offers viewed or the number of online applications submitted—has also grown. Understanding how these dimensions manifest in digital contexts has therefore become particularly relevant today.

By examining different operationalization strategies for this concept, the paper provides relevant insights for researchers who aim to improve the quality of the data they use when studying online behaviors. Understanding how seemingly small decisions in the operationalization of concepts can significantly impact outcomes is essential for advancing both methodological approaches and the interpretation of digital trace data in social science research.

## 2. Background

### 2.1 Operationalization of job search intensity

Job search intensity can be found in the literature operationalized in diverse ways (see Table 1 for a summary). Ellis et al. [24] provided evidence that job search intensity is a multidimensional construct, encompassing factors such as time and effort spent, timing of search initiation, and number of employers contacted. As a result, job search intensity can be operationalized in various ways, depending on how effort is measured and what time frames are considered. For example, Blau [25] used four items assessing general effort (e.g., "Devoted much effort to looking for a job.") rated on Likert-type scales, while Barber et al. [26] used multiple survey measures of job search intensity, including two multiple-choice items: one asking about hours spent job searching per week and another assessing subjective effort (rated on a 5-point scale from "very little effort" to "a great deal of effort"). Their pilot test revealed common challenges faced in survey-based measurements: respondents struggled to provide a specific numeric answer to the weekly hours question, often writing vague responses like "lots of hours." To address this, the authors used a scale with options ranging from "one hour or less" to "more than twenty hours." Subsequent studies on the same topic adopted similar approaches (e.g., [27,28]).

Researchers studying job search intensity in the era of Internet continued to rely on surveys, increasingly shifting toward online formats (e.g., [29,30]). In recent years, however, some have recognized the advantages of using digital trace data. Faberman and Kudlyak [31] utilized one year of proprietary web-tracking data from SnagAJob

**Table 1. Different operationalizations of job search intensity in the literature.**

| Authors | Data used | Operationalization |
| --- | --- | --- |
| Ellis et al. [24] | Paper survey | Multidimensional construct including time and effort spent, and number of employers contacted. |
| Blau et al. [25] | Paper survey (postal) | Four items assessing general effort (e.g., "Devoted much effort to looking for a job."), rated on a 5-point Likert scale ("Strongly Disagree" to "Strongly Agree"). |
| Barber et al. [26] | Paper survey (in person and postal) | Two different operationalizations were used:<br>1. Total number of job search sources used (0–8).<br>2. An operationalization based on two questions:<br> • Time spent, measured either (1) through an open question asking for hours per week, or (2) via a scale ranging from "one hour or less" to "more than twenty hours."<br> • Subjective effort, rated on a 5-point scale from "very little effort" to "a great deal of effort." |
| Saks and Ashforth [27] | Paper survey (postal) | Four items assessing general effort (e.g., "Spent a lot of time looking for job opportunities"), rated on a 5-point Likert scale ("Strongly Disagree" to "Strongly Agree"). |
| Wanberg et al. [28] Zacher [29] | Paper survey (in person and postal) | Seven items, including "used the Worldwide Web or other computer services to locate job openings". Response options ranged from 1 ("never or zero times") to 5 ("very often, at least ten times"). |
| da Motta Veiga [30] | Online survey | Four items assessing the extent to which participants used several tactics to find job openings (e.g., "used the internet to locate job openings"). Responses ranged from 1 ("very slightly or not at all") to 5 ("very frequently"). |
| Faberman and Kudlyak [31] | Proprietary web-tracking data | Number of applications during each job search spell, defined as groups of searches separated by more than five weeks of inactivity. |
| Ochoa and Revilla [21] | Metered data | Time spent per day on job search platforms. |

(www.snagajob.com), an online job search platform, to explore the relationship between search intensity and search duration. They found that, within a single search spell, search intensity consistently declined, challenging earlier labor search models which were mostly based on survey data, such as the one proposed by Pissarides [32].

Beyond defining and measuring job search intensity, past research has also employed several validation strategies to evaluate the reliability and validity of the measures. Internal consistency reliability has been frequently assessed using Cronbach's alpha or equivalent indices, with values above 0.70 indicating that items coherently capture the underlying construct [25,27–29]. Factor analytic techniques, such as confirmatory factor analysis (CFA), have been used to examine whether items reflect a single latent construct [28]. These techniques —widely used in social sciences, psychology, and other disciplines— require proposing and testing multiple items, ultimately selecting a subset that forms a composite score intended to measure the concept of interest. Consequently, assessing the reliability and validity of single items is limited, as these methods primarily evaluate the quality of the construct as a whole. Moreover, extending such approaches to metered data raises challenges, since the equivalents of survey items are behavioral metrics, each recorded on distinct and often incomparable scales (e.g., visits, durations, or counts).

Predictive and concurrent validity have also occasionally been assessed, with studies showing that higher job search intensity scores are associated with higher success in finding a job [25,27]. However, concurrent validity is based on associations measured at the same time, making it difficult to disentangle true relationships from shared-method variance. Predictive validity, in turn, is influenced by external factors that shape outcomes, meaning the evidence remains correlational and context-dependent. Furthermore, establishing similar forms of validity using metered data may be difficult when the predicted outcomes—such as employment success—cannot be directly observed within the digital traces themselves. This limitation could be addressed by complementing metered data with survey information collected from the same participants.

Together, these strategies illustrate the standard psychometric approach for establishing the soundness of survey-based measures of concepts such as job search intensity, as well as the methodological difficulties of adapting these approaches to metered data.

## 2.2 Operationalizing concepts using digital traces

Digital trace data are increasingly used to study a wide range of online behaviors and their societal implications. For instance, they have been employed to predict voter turnout based on online activity [33], examine Facebook's impact on the public agenda [34], analyze echo chambers in online news consumption [35], investigate how exposure to "fake news" shapes political attitudes [36], and explore the relationship between online vaccine-related information consumption, social media use, and attitudes toward vaccines [37]. These examples illustrate the growing methodological relevance of digital trace data across social and political sciences.

To the best of our knowledge, the only study using digital trace data to investigate job search behavior is Faberman and Kudlyak's [31] study, which highlights, Faberman and Kudlyak [31] the distinct challenges of operationalizing concepts, such as job search intensity, when using digital trace data rather than traditional survey data. First, the study relied on a specific type of digital trace data—web tracking data (i.e., users' activity recorded from the server side [38])—collected from a single platform primarily featuring hourly, lower-skilled jobs, which limited the generalizability of the results. This contrasts with the use of metered data, which can be collected across multiple platforms and provides a broader view of job search behavior. Second, the data they used could not capture when an individual started or ended their job search outside of the platform studied. This limitation is mitigated with metered data, as it allows for observing job searches across multiple platforms. However, searches conducted via alternative online methods (e.g., direct emails to companies) or offline methods (e.g., in-person applications), or through non-metered devices, remain unobservable.

Third, the authors faced several decisions when operationalizing job search intensity using the available data. They used the number of job applications recorded on the platform as the basic metric for measuring the concept, and defined individuals' job searches as distinct periods of activity, with each search spell separated by more than five weeks of inactivity. Consequently, job search intensity was defined as the number of applications submitted during each job search spell. Had these decisions been made differently, the results might have varied. The authors acknowledged that the number of applications may not be a perfect measure of search effort, as some individuals may spend considerable time on a few well-crafted applications, while others may apply to many jobs with little effort.

Ochoa and Revilla [21] used what they considered the most "promising" operationalization when measuring job search intensity through metered data: time spent on job search platforms per day, covering the entire period from the first to the last visit across platforms. They explored how decisions made before data collection—the type of devices tracked (PCs, mobile devices, or both), the specific time frame for data collection (e.g., the time of year), and the extension of the data collection period (e.g., one month, three months, or nine months)—affected the measures obtained. They found that limiting data collection to a single type of device, as well as varying the duration of data collection, could significantly impact outcomes.

Moreover, the authors acknowledged the range of alternative operationalizations that could have been applied using the available metered data. For instance, instead of time spent on job search platforms, other measures of job search activity could include the number of pages visited within platforms or the number of job offers viewed. If using a time-based metric, researchers must determine how to calculate the total duration of activity, as meters generally record access times to web pages or apps but not the duration of activity. One popular approach is to measure the time between the first page visit related to the activity and the next visit to an unrelated page, although other alternatives also exist. Additionally, any of these decisions depend on how researchers link the online activity of interest (e.g., job searching) to specific meter-recorded actions, including which websites and apps are considered relevant (e.g., list of job search platforms) and which specific pages or sections (e.g., URLs for job offer descriptions) should be observed.

Consequently, operationalizing concepts using metered data requires making multiple assumptions, which leads to diverse measures of the same concept. Bosch [39], building upon the multiverse analysis framework by Steegen et al. [40], explored 2,631 variations in operationalizing the concept "media exposure" to "hard news" (defined as news articles covering political, national, international, and regional affairs, as well as political opinion pieces [41]) using metered

data from Spain, Portugal, and Italy. He varied aspects such as the list of news websites included and the minimum time required to consider an individual "exposed" to a website. His results highlighted substantial fluctuations in measurements obtained for each operationalization and a low predictive validity, a data quality indicator that evaluates whether the measurements predict or correlate with an external criterion theoretically related to the construct being measured [42]. In this case, the external criterion used was political knowledge, assessed through four questions around basic knowledge about how institutions work and the composition of the current government. Using the same dataset, Bosch and Revilla [11] assessed the convergent validity of the measures. Convergent validity refers to the degree of correlation between independent measures of the same underlying concept [43]. If all operationalizations are effectively capturing the same concept, they should exhibit a high degree of correlation. However, Bosch and Revilla [11] found that only 26.4% of the correlations exceeded 0.70, indicating low convergent validity.

## 3. Research question and contribution

This study addresses the following question: How do different operationalizations of the concept "job search intensity" using metered data affect the outcomes?

In light of the existing but limited studies on this subject [11,39], we anticipate substantial variability in the results. This variability likely arises not only from the strength of the relationship between the available metered variables and the underlying concept (as with any operationalization), but also from the unique sources of error associated with metered data [44] and the decisions made during data processing and analysis.

The variability resulting from different operationalizations is assessed at three different levels: (1) the overall variability of measurements obtained across different operationalizations, (2) the variability arising from linking the concept to different metrics available through metered data that may capture job search intensity, such as the number of applications submitted or the time spent on job search platforms, and (3) the variability introduced by other operationalization factors, including the criteria used to segment job search activity into distinct search spells or the methods applied to address outliers. The practical implications of this variability are exemplified by comparing different measures of job search intensity across gender and age groups.

The study contributes to the existing literature in several ways. First, we build on the work of Bosch [39] by examining the variability introduced by operationalization decisions when using metered data in a new research context: job search. This allows us to assess whether the high variability observed in media studies also extends to other fields. Second, we explore additional operationalization factors and the potential values that can be assigned to these factors, beyond those considered in previous research. When compared with Ochoa and Revilla [21], we examine the effect of operationalization decisions made after data collection, and not those made during the planning stage (e.g., devices used, as well as the duration and timing of the data collection period). When compared with Bosch [39], we explore different methods for addressing outliers (e.g., adjustments for excessively long visit durations) and segmenting job search activity into distinct search spells.

Beyond its methodological contribution, this research aims to advance best practices in digital behavioral measurement by illustrating how specific operationalization choices can influence substantive conclusions. It should help researchers using metered data and other forms of digital trace data better understand the limits and robustness of their results. Moreover, by identifying the main sources of variability and uncertainty, this study seeks to motivate future research to determine which operationalization factors are most critical for ensuring valid and comparable measures across studies.

The implications of this work extend beyond the study of job search behavior. As digital trace data are increasingly used across diverse disciplines—such as labor economics, public policy, communication studies, marketing, education, and computational social science—the insights provided here can help scholars in these fields develop more transparent, replicable, and theoretically grounded approaches to operationalizing behavioral constructs in digital environments.

## 4. Data and methods

### 4.1 Ethics and data protection

This study involved human participants' data and was approved by the Institutional Committee for Ethical Review of Projects (CIREP) at Pompeu Fabra University (Approval No. 135). All ethics approval documents were issued in English. The data analyzed were anonymized by Netquest before being shared with the research team and included only visits to job search platforms, with no personally identifiable information accessible. Participants' consent was obtained by Netquest through its metered panel procedures, which include explicit consent both when joining the panel and when installing the Wakoopa meter, a tracking software developed by Netquest. The project also received approval from an external ethics advisor, in accordance with European Research Council requirements, who supervised data anonymization and compliance with ethical standards. Supporting ethics and data protection documents are available in SOM1.

### 4.2 Data

The data used in this research were sourced from the Netquest metered panel, which consists of individuals who regularly participate in surveys and have also installed the Wakoopa meter on some or all of their browsing devices. The analyzed dataset is a subset of these metered data, collected from both PCs and mobile devices in Spain, from the 1st of March to the 30th of November 2023 (a nine-month period). The dataset was accessed for research purposes on 18/4/2024. This dataset includes records of visits to specific online job search platforms (see SOM2), encompassing both websites and mobile applications (apps). The panel company did not provide any information that could identify individual participants, and the metered data used were limited to visits to job search platforms, preventing the inference of participant identities.

The final dataset comprises 600 panelists, selected from a pool of 18,980 metered panelists in Spain. The selection criteria included providing metered data throughout the entire study period, having visited at least one of the job search platforms of interest on at least five different days, and having previously supplied demographic information on gender, age, region, and social class. From this eligible group, 600 panelists were randomly selected.

Metered data were combined with 53 panel profiling variables (not used in this study; see SOM3), including the four sociodemographic variables used as selection criteria. The average age of participants is 41 years (compared to 42 in the entire metered panel), with 53% being women (50% in the panel). 51% are mid-educated (50% in the panel), and 47% highly educated (40% in the panel). Of the observed panelists, 73% shared metered data from both PC and mobile devices (smartphones and tablets), 22.0% from mobile devices only and 4.8% from PCs only. The total number of pages viewed on job search websites and app visits is 209,518, averaging 349.2 pages/app visits per panelist over the nine-month period.

### 4.3 Multiple operationalizations of job search intensity

Following Bosch [39], we begin by identifying the metrics available in the metered data that can be used to operationalize job search intensity. These metrics are constructed by combining two **measurement types**—time spent and number of visits—with four **measurement targets**: browsing sessions on job search platforms, pages within platforms, specific pages presenting job offers, and application pages.

A **browsing session** refers to a continuous period of activity on a job search platform. For example, a panelist visiting InfoJobs.net (the leading job search platform in Spain) and navigating through various sections of the site without significant interruptions (e.g., within the same 30-minute window) would be counted as a single browsing session. A page within the platform refers to any individual web page visited. In the case of InfoJobs.net, this could include the homepage, search results pages, or company profile pages. A specific page presenting a job offer is a page displaying details about a particular job listing. For instance, if the panelist clicks on a job post for a "Marketing Specialist" position, the detailed job description page would be counted under this category. An application page refers to a web page where the user initiates

or completes a job application. On InfoJobs.net, this could be the page where the panelist uploads their résumé, fills out an application form, or clicks the "Apply" button to submit their interest in a specific job. The distinction between generic pages (e.g., the platform homepage or search results) and more specific pages, such as job offer or application pages, is relevant as they capture different aspects of the job search process.

This approach results in eight distinct metrics (see Table 2), some of which have been used in previous research, like the time spent on job search platforms [21] or the number of applications [31], which we approximate here by counting visits to application pages. By systematically combining measurement types and targets, we introduce new potential metrics, some of which may serve as particularly strong indicators of job search intensity, such as the number of job offer pages visited.

Each of these metrics can be calculated in different ways, depending on several **operationalization factors**. This study focuses on the decisions researchers must make when working with data that has already been collected. For an analysis of the impact of pre-collection decisions, see Ochoa and Revilla [21].

Table 3 presents the operationalization factors and their explored values, which are used to create multiple operationalizations of "job search intensity" in combination with the eight metrics outlined in Table 2.

The first factor considered is **platform type**. Depending on the devices used for data collection, researchers may have access to job search activity on websites, apps, or both. However, the level of detail available for each platform type

**Table 2. Metrics used to operationalize "job search intensity".**

| Metric | Measurement type | Measurement target |
|---|---|---|
| **M1**: Time spent on browsing sessions on job search platforms | Time | Platforms |
| **M2**: Time spent visiting pages within platforms | Time | Platform pages |
| **M3**: Time spent visiting job offers pages within platforms | Time | Job offers pages |
| **M4**: Time spent visiting application pages within platforms | Time | Application pages |
| **M5**: Number of browsing sessions on job search platforms | Visits | Platforms |
| **M6**: Number of pages visited within platforms | Visits | Platform pages |
| **M7**: Number of job offers pages visited within platforms | Visits | Job offers pages |
| **M8**: Number of application confirmation pages visited (used as a proxy for the number of applications submitted) | Visits | Application pages |

**Table 3. Operationalization factors of "job search intensity".**

| Factor | Description | Values |
|---|---|---|
| Platform type | Types of platforms considered | Websites, apps, and both *Reference: both* |
| Search spell separation time | Minimum time between job search activities to consider them as separate search spells | No minimum, 30, 60, and 90 days. *Reference: No minimum* |
| Session time | Minimum time between job search activities to consider them as separate browsing sessions | 10, 30, and 60 minutes. *Reference: 10 minutes* |
| Minimum time | Minimum time spent on a page or session for it to be considered valid | No minimum, 2, 5, and 10 seconds. *Reference: No minimum* |
| Maximum time | Maximum time spent on a page visit before it is considered implausible | No maximum, 10, 20, and 30 minutes. *Reference: No maximum* |
| Maximum time outlier treatment | Method used to handle maximum time outliers | (1) Removal, (2) Replacement with maximum, (3) Replacement with mean, (4) Replacement with median. *Reference: Removal* |

depends on the meter used and the panelists' devices. For instance, the meter used in this study records all four measurement targets considered in this research for websites but only captures platform visits (sessions) for apps. Therefore, only metrics M1 and M5, as defined in Table 2, can be assessed with app data. Moreover, even when using metrics that can be computed for both web and app data, researchers may hesitate to combine the two due to concerns about the comparability of these metrics across platform types. For example, completing the same job search tasks may systematically take longer on apps than on websites, as app users rely exclusively on mobile devices, where typing on a touchscreen keyboard is more challenging than on a PC.

Next, the criteria used to determine when a job search spell begins and ends directly impact measures of job search intensity. For example, if a panelist engages in job search activity during two separate periods with one month of inactivity in between, these periods can either be classified as two distinct job search spells or as a single continuous spell. The minimum inactivity period (**search spell separation time**) required to mark the end of one search spell and the start of another affects all job search intensity metrics, as the inclusion or exclusion of inactive periods influences the calculations. Additionally, when a panelist's job search activity is divided into multiple spells, they contribute multiple observations of job search intensity, potentially impacting the overall measures.

Within each search spell, job search activity on job search platforms can be measured at two levels: page level (each individual platform page visited) and session level. A session is defined as a sequence of consecutive page visits, with no more than a specified **session time** gap between them. The choice of this time gap affects both the number and duration of sessions recorded for the same browsing activity. Metrics M1 and M5 are session-based rather than page-based, meaning their values depend on how sessions are defined.

The treatment of time outliers also impacts the operationalization of job search intensity. For instance, visits—whether to platforms or individual pages—can be filtered based on **minimum time** thresholds to exclude those deemed too short for meaningful user interaction. Similarly, **maximum time** thresholds can be applied to adjust for excessively long visit durations, which may indicate user inactivity (e.g., switching to unrelated tasks while the page remains open). While extremely short visits are typically removed, there are different **maximum time outlier treatments** for handling excessively long visit times: options include discarding the visit entirely, capping the recorded time at the maximum considered an outlier, or replacing it with the mean or median time of the remaining observations [45]. These adjustments are more complex and less effective for app data, as they must be applied at the session level rather than the page level. In this study, we applied the same minimum time thresholds to app visits and to page visits, acknowledging that, in practice, only short single-page app visits would be classified as outliers. However, we did not apply maximum time thresholds to app visits, as these visits correspond to sessions, and job search sessions can be prolonged without necessarily indicating user inactivity.

The combination of metrics and operationalization factors results in 1,296 measures for session-based metrics (M1 and M5) and 1,248 for page-based metrics (all others). The difference arises because 48 combinations involve setting the device type factor to app data only, which prevents the calculation of page-based metrics.

Finally, the way metrics are summarized across individuals in the sample—using **summary statistics** such as the median versus the mean—can influence the conclusions drawn from the data. For instance, the median is often more robust to outliers than the mean. To explore these effects, the individual measures obtained by combining metrics and operationalization factors are summarized across individuals using both the mean and the median, resulting in a total of 2,592 sample measures for session-based metrics and 2,496 sample measures for page-based metrics.

The levels of operationalization factors were chosen to reflect prior literature or common practices where possible, while also exploring plausible variation. For instance, search spell separation time was defined around the five-week threshold used by Faberman and Kudlyak [31], with values of 0, 30, 60, and 90 days to test sensitivity. Session time follows the 30-minute standard used in Google Analytics that is also commonly applied in studies using metered data

(e.g., [46]), with 10- and 60-minute alternatives used to capture shorter or longer sessions. Minimum time thresholds (2, 5, and 10 seconds) were inspired by Bosch [39], adapting page-level exposure thresholds to the job search context, where brief visits may suffice to evaluate a job posting. For other factors, plausible values were selected based on expected user behavior.

Fig 1 summarizes the process through which multiple operationalizations of job search intensity are derived by combining measurement types, measurement targets, and operationalization factors.

## 4.4 Analyses

The operationalization of variables and subsequent analyses were performed using R (version 4.2.3) [47]. The variability resulting from different operationalizations is then analyzed from multiple perspectives.

First, following Bosch and Revilla [11], we assess the overall variability of the operationalizations by examining the extent to which these measures correlate with each other. The infeasibility of determining the 'true' value of job search intensity for the sample (i.e., the actual effort individuals exerted in searching for a job) or a closely associated measure (e.g., obtained through a survey conducted on the same individuals) prevents us from identifying the most valid operationalization. Instead, the average and median correlations among measurements are used as indicators of convergent validity (similar to Bosch and Revilla [11]).

With 1,296 operationalizations for metrics M1 and M5 and 1,248 for the remaining six metrics, the total number of distinct measurements amounts to 10,080, with operationalizations based on the same metric using the same scales. These measurements lead to 50,798,160 correlations between pairs of measurements. The variability of these measurements is assessed by presenting the distribution of correlations through a histogram and calculating the interquartile range (the difference between the first and third quartiles), which covers the central 50% of correlations.

Second, the convergent validity between the eight metrics is assessed by computing the average correlation between all measurements using each pair of metrics. If two metrics exhibit a high pairwise average correlation, this would suggest that both are measuring the same underlying concept. Conversely, if low pairwise average correlations are observed, it would indicate that some metrics may not be properly measuring the concept of interest, even though it would not reveal which specific metric is problematic. This would underscore the importance of metric choice in the operationalization process, along with other operationalization factors.

Third, we assess how operationalization factors influence variability within each metric used to measure job search intensity and how the use of different summary statistics to aggregate these measurements across individuals impacts the results.We quantify how operationalization factors affect measurements within each metric by computing correlations between measurements sharing the same metric, where lower correlations indicate larger effects.. To evaluate the effect of the summary statistics, we calculate both the mean and the median of individual measurements for each operationalization within each metric. We then assess the variation of these summarized measurements across operationalizations that share the same metric by calculating the standard deviation for both the means and medians. Because only measurements based on the same metric share the same scale, we normalize the standard deviations by dividing them by the mean (whether of the means or medians) to obtain the coefficient of variation. This normalization ensures fair comparisons among metrics.

Fourth, to determine which operationalization factors have the greatest impact on the variation in the measures obtained for each metric, eight mixed-effects linear regressions are conducted. In these models, the dependent variable is the measure derived from each metric, while the operationalization factors serve as independent variables. Random effects are included for individual panelists, as each contributes multiple observations from different search spells across various platform types. Model fit is assessed using the Akaike Information Criterion (AIC), Bayesian Information Criterion (BIC), and Root Mean Square Error (RMSE), which indicate the overall adequacy and predictive accuracy of each model.

```
Start
  ↓
Identify available metered data
  ↓
Select Measurement Type
  ├── Time spent
  └── Number of visits
                ↓
Select Measurement Target
  ├── Browsing sessions on job search platforms
  ├── Pages within platforms
  ├── Job offer pages
  └── Application pages
                ↓
Combine Measurement Type + Target
  → 8 Core Metrics (M1–M8)
                ↓
Apply Operationalization Factors
  ├── ... to M1 and M5 (session-based metrics)
  │       ├── Platform type: web / app / both
  │       ├── Search spell separation time: no min / 30 / 60 / 90 days
  │       ├── Session time: 10 / 30 / 60 min
  │       ├── Minimum time: no min / 2 / 5 / 10 sec
  │       ├── Maximum time:
  │       │       ├── No maximum
  │       │       └── Apply maximum threshold (10 / 20 / 30 min)
  │       │               ↓
  │       │           Maximum time outlier treatment (only if max applied)
  │       │                   ├── Removal
  │       │                   ├── Replacement with maximum
  │       │                   ├── Replacement with mean
  │       │                   └── Replacement with median
  │       ↓
  │       Compute metric for each combination → 1,296 measures
  │
  └── ... to M2–M4 and M6–M8 (page-based metrics)
          ├── Platform type: web
          ├── Search spell separation time: no min / 30 / 60 / 90 days
          ├── Minimum time: no min / 2 / 5 / 10 sec
          ├── Maximum time:
          │       ├── No maximum
          │       └── Apply maximum threshold (10 / 20 / 30 min)
          │               ↓
          │           Maximum time outlier treatment (only if max applied)
          │                   ├── Removal
          │                   ├── Replacement with maximum
          │                   ├── Replacement with mean
          │                   └── Replacement with median
          ↓
          Compute metric for each combination → 1,248 measures
                ↓
Summarize Across Individuals
  ├── Mean
  └── Median
                ↓
Final Output
  → 2,592 session-based measures (M1, M5)
  → 2,496 page-based measures (M2–M4, M6–M8)
```

**Fig 1. Process of constructing multiple operationalizations of job search intensity using metered data.**

Since the eight metrics operate on different, non-comparable scales, they are standardized (i.e., mean-centered and divided by the standard deviation) to enable comparisons of the relative importance of each operationalization factor. This standardization ensures that each regression coefficient represents the expected change in the measure—relative to the reference level of the corresponding operationalization factor—expressed in standard deviation units. As a result, regression coefficients can be directly compared across metrics.

Finally, to illustrate the practical implications of the variability caused by different operationalizations, job search intensity is compared across gender (males vs. females) and age groups (<45 vs. ≥ 45 years old), following the analysis performed by Ochoa and Revilla [21]. For each metric, the mean of the multiple standardized measures is compared between groups. Additionally, t-tests are conducted to compare the means between groups for each operationalization within a metric, assessing the proportion of operationalizations that yield statistically significant differences.

## 5. Results

### 5.1 Overall variation

Fig 2 presents the distribution of correlations between all pairs of measurements.

Correlation variability is substantial: 50% of the correlations fall between 0.22 and 0.71 (first and third quartiles), with values ranging from −1 to +1. This high variability is consistent with Bosch [39] and Bosch and Revilla [11], who reported a central 50% interval between 0.15 and 0.63, when studying news media consumption using metered data in Spain.

The average correlation is 0.44, and the median 0.46. Additionally, only 26.4% of the correlations exceed 0.70, indicating low convergent validity among the various operationalizations explored. This result also aligns with Bosch and Revilla [11], who reported a mean correlation of 0.40 and a median of 0.41.

### 5.2 Variation between metrics

Fig 3 illustrates the average correlation between measurements derived from each pair of metrics defined in Table 2.

The results show that while some metrics produce high average correlations between their measurements (e.g., M2-M6: 0.91), others are almost uncorrelated (e.g., M5-M8: 0.14). A correlation of 0.14 indicates virtually no linear relationship, meaning high values on one metric do not predict high values on the other; consequently, using one metric or the

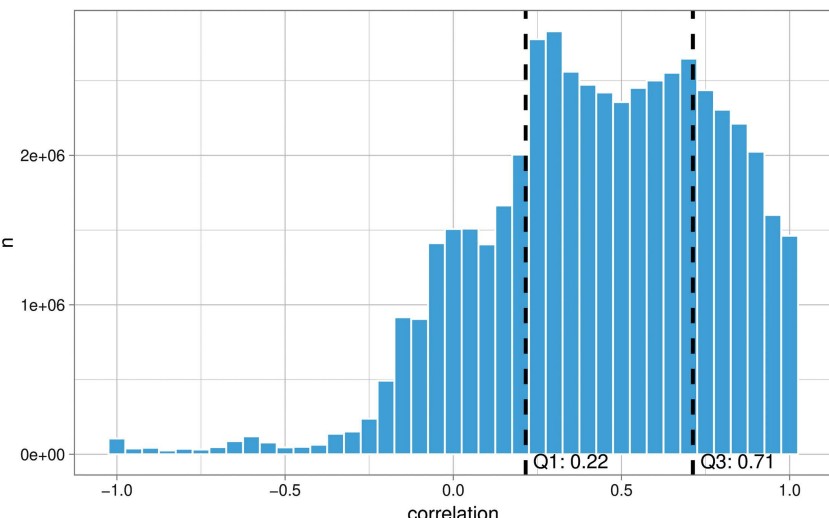

**Fig 2. Distribution of correlations between pairs of measurements.**

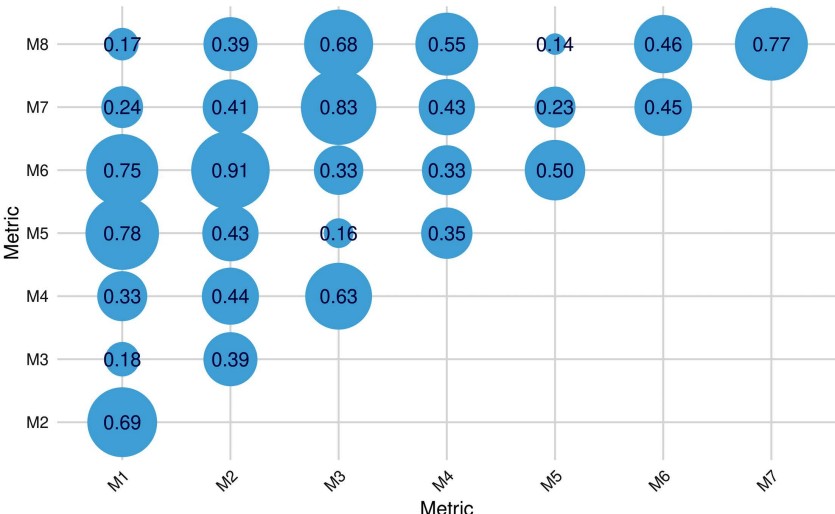

**Fig 3. Average correlation between measurements obtained for each pair of metrics.**

other could lead to opposite conclusions in practice. In contrast, a correlation of 0.91 suggests that the metrics capture very similar patterns of job search activity.

In general, metrics that share the same measurement target (M1-M5, M2-M6, M3-M7, and M4-M8) exhibit stronger average correlations than those that share the same measurement type (M1-M2-M3-M4 and M5-M6-M7-M8). This suggests that selecting the event that most accurately reflects the concept of interest (e.g., sessions, visited pages, etc.) may be more critical than choosing whether to measure event counts or durations. Additionally, the presence of very low average correlations suggests that not all metrics are effectively measuring the concept of interest, highlighting the relevance of carefully selecting the most appropriate metric.

### 5.3 Variation within metrics

Next, we examine the variation among operationalizations based on each metric defined in Table 2, resulting from the combination of operationalization factors listed in Table 3. For each metric, we compute all possible measures per panelist by applying different combinations of operationalization factors.

Table 4 presents the average correlation (column "Avg. Corr.") between all pairs of measurements (with the total number of measurements per metric indicated in column "n") within the same metric. Additionally, Table 4 illustrates how the results vary when both the mean and the median are used to summarize measurements across individuals. To assess this variation, the standard deviation (SD) of these measures is presented, along with the corresponding coefficient of variation (coef. var.), which is calculated by normalizing the SD by the mean (computed from either the mean or the median of the measures). This approach ensures a fair comparison of variability across metrics that may have different units and scales.

In general, the average correlation within each metric (ranging from 0.53 to 0.81) is larger than the average correlation between metrics (see Fig 3; correlations ranging from 0.14 to 0.91). This reinforces the idea that the choice of metric to operationalize a concept has a greater impact on the results than the operationalization factors explored in this study.

When summarizing a metric using the mean, the coefficient of variation ranges from 24.5% for M2 (time spent visiting job offers pages per day) to 83.9% for M4 (time spent on application processes per day). This suggests that (1) variation due to operationalization decisions—beyond the choice of metric—is substantial, and (2) some metrics are more robust to these decisions than others.

**Table 4. Variation in measurements for each metric due to operationalization factors.**

| Metric | Units | n | Avg. Cor. | Summary statistic = mean | | | Summary statistic = median | | |
|--------|-------|---|-----------|------|------|----------|------|------|----------|
| | | | | SD | Mean | coef. var. (%) | SD | mean | coef. var. (%) |
| M1 | time session (sec.)/ day | 1,296 | 0.59 | 51.3 | 139.5 | 36.8 | 11.8 | 35.9 | 32.9 |
| M2 | time pages (sec.)/ day | 1,248 | 0.80 | 15.2 | 62.4 | 24.5 | 6.7 | 18.0 | 37.1 |
| M3 | time job offers (sec.)/ day | 1,248 | 0.81 | 11.0 | 29.1 | 37.9 | 5.6 | 8.2 | 68.6 |
| M4 | time applications (sec.)/ day | 1,248 | 0.54 | 11.8 | 14.1 | 83.9 | 2.3 | 3.7 | 60.9 |
| M5 | sessions/ day | 1,296 | 0.53 | 0.13 | 0.44 | 30.0 | 0.11 | 0.23 | 50.0 |
| M6 | pages/ day | 1,248 | 0.78 | 0.64 | 2.02 | 31.6 | 0.26 | 0.67 | 38.2 |
| M7 | job offers/ day | 1,248 | 0.80 | 0.35 | 0.86 | 40.8 | 0.28 | 0.33 | 84.1 |
| M8 | applications/ day | 1,248 | 0.72 | 0.18 | 0.26 | 69.3 | 0.06 | 0.07 | 83.8 |
| Average: | | | 0.69 | | | 44.4 | | | 57.0 |

Using the median instead of the mean consistently yields lower estimates, ranging from 25.7% of the mean-based estimate for M1 (35.9/139.5) to 52.5% for M5 (0.23/0.44). This large discrepancy suggests that all metrics have strongly right-skewed distributions, with high-value outliers affecting the mean more than the median.

Moreover, summarizing with the median generally results in greater variation: six out of eight metrics show higher coefficients of variation when using the median compared to the mean. On average, the coefficient of variation across metrics is 44% when using the mean, compared to 57% with the median. This pattern suggests that the median may be more sensitive to operationalization choices.

Overall, metrics M2 and M3 (time spent on pages and job offer pages) show the highest average correlations and relatively low variation across operationalizations, suggesting that they are more robust indicators of job search intensity. In contrast, metrics M4, M7, and M8 (time on applications, job offers, and applications counts) show higher variability, indicating greater sensitivity to operationalization choices.

## 5.4 Effect of operationalization factors on measurements for each metric

Table 5 presents the results of regression analyses assessing the impact of each operationalization factor on the final standardized measures. The table reports the estimated effects of each factor level relative to its corresponding reference level (see Table 3) and indicates statistical significance ($p < 0.05$).

Since page-based metrics (all except M1 and M5) cannot be computed when restricting data to app usage, coefficients for "Platform type: app" are not reported. Additionally, while all other factors were applied to every metric, some did not introduce any variation in the final measurements, resulting in null coefficients. Specifically, session time does not affect page-based metrics, while maximum time and maximum time treatment do not impact metrics based on counts of page visits (M6, M7, and M8).

Overall, most factor levels in the regressions (62 out of 98) show significant effects. Platform type has a moderate but consistent influence, with 9 out of 10 coefficients being significant, though the largest effect reaches only 0.095 standard deviation units. The strongest effects come from search spell separation time, where applying a 30-day threshold results in changes between 0.208 and 0.646 standard deviation units, with all levels of this factor significantly affecting the metrics. Session separation time also plays a key role, with all coefficients significant and a maximum effect of 0.220 standard deviation units.

Corrections for minimum time per visit have a more substantial impact than those for maximum time, with 20 out of 24 coefficients being significant and a maximum effect of 0.228 standard deviation units for M7. In contrast, maximum time corrections show only 6 significant effects out of 15, with the largest effect at 0.045 standard deviation units. The method used for outlier correction does not produce significant differences.

**Table 5. Impact of operationalization factors on measurements for each metric.**

| Factor | Level | M1 | M2 | M3 | M4 | M5 | M6 | M7 | M8 |
|---|---|---|---|---|---|---|---|---|---|
| Platform type | web | −0.029* | 0.033* | 0.069* | 0.073* | −0.003 | 0.031* | 0.04* | 0.029* |
| | app | 0.013* | | | | 0.095* | | | |
| Search spell separation time | 30 days | 0.208* | 0.300* | 0.377* | 0.517* | 0.646* | 0.303* | 0.378* | 0.428* |
| | 60 days | 0.119* | 0.169* | 0.193* | 0.078* | 0.318* | 0.167* | 0.191* | 0.334* |
| | 90 days | 0.067* | 0.097* | 0.123* | 0.026* | 0.173* | 0.093* | 0.101* | 0.105* |
| Session time | 30 min. | 0.088* | 0.000 | 0.000 | 0.000 | −0.072* | 0.000 | 0.000 | 0.000 |
| | 60 min. | 0.220* | 0.000 | 0.000 | 0.000 | −0.113* | 0.000 | 0.000 | 0.000 |
| Minimum time | 2 sec. | −0.027* | 0.019* | 0.004 | −0.001 | 0.003 | −0.080* | −0.094* | −0.025* |
| | 5 sec. | −0.050* | 0.021* | 0.042* | −0.013 | −0.013* | −0.163* | −0.164* | −0.136* |
| | 10 sec. | −0.064* | 0.036* | 0.041* | −0.072* | −0.007* | −0.230* | −0.228* | −0.204* |
| Maximum time | 30 min. | −0.004 | −0.008* | 0.001 | −0.038* | 0.003 | 0.000 | 0.000 | 0.000 |
| | 20 min. | −0.005 | −0.009* | 0.001 | −0.041* | 0.002 | 0.000 | 0.000 | 0.000 |
| | 10 min. | −0.011* | −0.017* | −0.001 | −0.045* | 0.003 | 0.000 | 0.000 | 0.000 |
| Maximum time treatment | threshold | 0.002 | 0.001 | 0.000 | 0.011 | −0.003 | 0.000 | 0.000 | 0.000 |
| | mean | −0.001 | −0.004 | −0.001 | −0.011 | −0.003 | 0.000 | 0.000 | 0.000 |
| | median | −0.001 | −0.004 | −0.001 | −0.012 | −0.003 | 0.000 | 0.000 | 0.000 |
| (Intercept) | | −0.146* | −0.177* | −0.267* | −0.232* | −0.272* | −0.052 | −0.074 | −0.102 |
| AIC (×10³) | | 2,497.1 | 2,064.2 | 221.4 | 99.7 | 2,428.6 | 2,118.5 | 237.1 | 41.0 |
| BIC (×10³) | | 2,497.3 | 2,064.5 | 222.6 | 99.9 | 2,428.8 | 2,118.7 | 237.2 | 41.1 |
| RMSE | | 0.81 | 0.73 | 0.59 | 0.77 | 0.78 | 0.75 | 0.62 | 0.65 |

*Reference levels for each factor used as explanatory variable can be found in Table 3. * $p < 0.05$.*

Overall, search spell and session separation times have the most substantial effects on job search intensity measurements, while platform type and maximum time corrections have a more limited influence.

## 5.5 Practical application

Finally, to illustrate how operationalization choices affect substantive findings, Tables 6 and 7 compare job search intensity by gender and age group (<45 vs. ≥ 45), using the average of all possible operationalizations for each metric. In both tables, the column "Diff" displays the difference between groups after standardizing the measurements (columns indicating "std"), while the column "% of sign. diff." reports the proportion of operationalizations that yielded significant differences.

The average measure obtained using six out of the eight metrics shows lower job search intensity among females. However, depending on the metric and operationalization factors used, such differences may or may not be significant. For example, while 16.7% of the operationalizations based on metric M1 show significant gender differences, none of the operationalizations based on metrics M4 and M6 produce significant differences. For age groups, all metrics indicate lower job search intensity among the older group, but again, the proportion of operationalizations per metric that yield significant effects varies considerably (18.8% for M7, versus 0% for M2, M3, and M6).

## 6. Discussion

### 6.1 Summary of main results

These results reveal substantial variability in measurements of job search intensity across the different operationalizations explored, as evidenced by the wide dispersion of correlations between measurement pairs. The low average pairwise

**Table 6. Comparison of average job search intensity per gender.**

| Metric | Units | Avg. Males | Avg. Females | Avg. Males (std) | Avg. Females (std) | Diff. (std) | % of sign. diff. |
|--------|-------|-----------|-------------|------------------|--------------------|-------------|------------------|
| M1 | time session (sec.)/ day | 164.4 | 125.0 | 0.040 | −0.049 | −0.089 | 16.7 |
| M2 | time pages (sec.)/ day | 68.4 | 55.2 | 0.011 | −0.069 | −0.080 | 1.4 |
| M3 | time job offers (sec.)/ day | 36.5 | 23.0 | 0.034 | −0.177 | −0.212 | 3.4 |
| M4 | time applications (sec.)/ day | 11.1 | 16.8 | −0.150 | −0.035 | 0.115 | 0.0 |
| M5 | sessions/ day | 0.45 | 0.40 | −0.038 | −0.122 | −0.084 | 12.4 |
| M6 | pages/ day | 2.23 | 1.79 | 0.013 | −0.071 | −0.085 | 0.0 |
| M7 | job offers/ day | 1.05 | 0.70 | 0.023 | −0.182 | −0.204 | 9.4 |
| M8 | applications/ day | 0.21 | 0.29 | −0.180 | −0.036 | 0.144 | 3.1 |

**Table 7. Comparison of average job search intensity per age group.**

| Metric | Units | Avg. <45 | Avg. ≥45 | Avg. <45 (std) | Avg. ≥45 (std) | Diff. (std) | % of sign. diff. |
|--------|-------|----------|----------|----------------|----------------|-------------|------------------|
| M1 | time session (sec.)/ day | 146.9 | 139.0 | 0.000 | −0.017 | 0.018 | 0.1 |
| M2 | time pages (sec.)/ day | 62.2 | 60.9 | −0.027 | −0.035 | 0.008 | 0.0 |
| M3 | time job offers (sec.)/ day | 32.4 | 26.3 | −0.030 | −0.126 | 0.096 | 0.0 |
| M4 | time applications (sec.)/ day | 16.7 | 10.1 | −0.037 | −0.169 | 0.132 | 0.5 |
| M5 | sessions/ day | 0.44 | 0.41 | −0.058 | −0.112 | 0.054 | 7.6 |
| M6 | pages/ day | 2.10 | 1.90 | −0.013 | −0.052 | 0.039 | 0.0 |
| M7 | job offers/ day | 1.05 | 0.67 | 0.020 | −0.200 | 0.220 | 18.8 |
| M8 | applications/ day | 0.30 | 0.22 | −0.008 | −0.164 | 0.156 | 0.0 |

correlation highlights limited convergent validity and suggests that some operationalizations may not effectively measure job search intensity.

A detailed analysis indicates that metric choice has the greatest impact on variability. Measurements based on different metrics show average correlations ranging from 0.14 to 0.91, whereas those using the same metric exhibit higher correlations (0.53 to 0.81), reflecting greater convergent validity. Metrics targeting the same measurement objective (e.g., sessions, web pages, job offer pages, or application pages) tend to correlate more strongly than metrics that share the same measurement type (time or visits), emphasizing the importance of selecting the appropriate target.

Although other operationalization factors have a smaller impact compared to metrics, their influence is still notable. When using the mean to summarize measurements sharing the same metric across individuals, the coefficient of variation of the means ranges from 24.5% to 83.9%, with some metrics being more sensitive to the selection of operationalization factors. Using the median instead of the mean slightly increases these coefficients, which range from 32.9% to 84.1%. Regression analyses show that search spell separation time, session separation time, and minimum visit duration significantly affect standardized measurements. Among these, search spell separation time has the strongest effect, with changes ranging from 0.208 to 0.646 standard deviation units for a 30-day threshold. In contrast, platform type and maximum time corrections exhibit smaller and less consistent effects.

Importantly, these variations have practical implications for conclusions in substantive studies. For example, comparisons by gender show that most metrics suggest lower job search intensity among females, but the proportion of operationalizations producing statistically significant differences varies widely: 16.7% for M1 versus 0% for M4 and M6. Similarly, for age groups, older individuals generally exhibit lower search intensity, yet significance varies by metric (18.8% for M7

versus 0% for M2, M3, and M6). This illustrates that both the choice of metric and operationalization factors can influence not only the magnitude but also the statistical significance of observed differences, underscoring the need for careful metric selection in applied research.

## 6.2 Limitations

This study is based on data from a single online panel (Netquest) in one country (Spain) and relies on specific metering technology. Consequently, the observed variability and robustness of job search intensity metrics may not generalize to other countries, labor markets, or cultural contexts, where online job search behavior and platform usage may differ. Moreover, studies addressing other research topics using metered data may exhibit different degrees of variability.

The capabilities of the metering technology are particularly significant. Different meters vary in their ability to capture search terms, mobile web visits, individual web pages, app usage, within-app pages, and web/app page content. These differences directly affect the available operationalization options. Although in this study we have focused on the operationalization decisions that must be made after collecting the data, such decisions are conditioned by the specific data captured by the meter. For instance, the inability of the meter used in this study to detect within-app pages meant that app data could not be integrated into most operationalizations. Findings may therefore differ with meters that provide broader coverage or more granular data.

Additionally, while this study examines how different operationalizations produce varying measurements and assesses their correlations as evidence of convergent validity, it does not evaluate other forms of validity, such as predictive validity, which could help assess which operationalizations better capture the concept of interest. Predictive validity could be examined by complementing metered data with survey data from the same individuals—an approach already employed by Bosch [30]. This is feasible when using metered panels built on regular survey panels, where participants provide both data types.

Future research could also replicate this analysis in other countries or labor markets, explore different metering technologies, and investigate how operationalization choices affect relationships with external variables such as employment outcomes, job offers, or job seeker well-being. Moreover, the same approach could be applied to other research topics to assess whether similar conclusions about metric variability and sensitivity to operationalization decisions hold.

## 6.3 Practical implications

The low convergent validity observed in this study mirrors findings by Bosch [39] and Bosch and Revilla [11] on measuring media exposure, which also revealed similar variability and low average pairwise measurement correlations. This consistency suggests that low convergent validity could extend to other concepts, being a general issue when using metered data. However, further research is needed to investigate this potential limitation using diverse operationalizations on metered panels.

Assuming that at least one of the operationalizations used in this study effectively measures job search intensity, the low convergent validity—particularly for some pairs of metrics—suggests that certain operationalizations may have low predictive validity. This finding serves as a caution to researchers who assume that metered data—and digital traces in general—offer inherently superior data quality. Although digital traces have clear advantages over survey data, the operationalization process remains essential. The best operationalization should be selected based on its ability to produce valid measurements of the concept of interest, incorporating complementary data sources when necessary (e.g., surveys).

Moreover, while metric selection is the most critical decision in the operationalization process, other operationalization factors—such as search spell separation time, session separation time, and minimum visit duration—can also significantly impact results. Some findings may challenge intuition; for example, corrections for maximum visit duration have less influence than expected, as extremely short visits are more common and therefore more impactful.

More research is needed to establish a set of best practices for applied research when multiple plausible operationalizations are available. Such practices could include the following:

- Ground operationalizations in theory and domain knowledge: Using knowledge about the topic can improve metric selection. Operationalizations should be theoretically justified and reflect meaningful effort. For example, time spent reading detailed job offers may better capture active job search than total pages viewed, while viewing job offers in general may reflect general interest in the labor market rather than genuine job search. Submitting applications may indicate actual job search effort.

- Consider data availability: Even if some events better capture the concept of interest than others (e.g., applications better capture effort than viewing job offers), their observability among sampled individuals may limit their usefulness. For instance, the meter used in this study could not track within-app pages, restricting certain operationalizations.

- Anticipate potential biases: Evaluate how metrics may over- or under-estimate behavior. For example, counting job offers viewed may overstate search activity if users browse casually, while counting sessions may inflate effort for users who log in frequently but engage superficially.

- Account for operationalization factors and data corrections: Adjust thresholds and handle outliers according to the data and research question. Unusually long page durations may reflect inactivity rather than effort, making outlier correction more relevant for time-based metrics. Similarly, brief visits made only to check the labor market can be filtered by setting appropriate minimum durations per visit.

- Validate with external data when possible: Comparing metered measures with survey data or other independent sources can help assess predictive or external validity and identify which metrics most accurately reflect the concept of interest.

- Assess robustness: Computing multiple operationalizations, even using a limited set of options, can help evaluate the stability of results. As illustrated in this study, conclusions about gender or age differences in job search intensity may vary depending on the metric and operationalization used.

All in all, successfully operationalizing concepts using digital trace data is a process that requires substantive knowledge of the topic, an understanding of how digital traces are collected, and thorough testing of different options. In this sense, a clear advantage compared to the operationalization process in surveys is that most of the decisions and their effects can be tested on the same collected data. While in surveys the number of options to operationalize a concept must be limited to a few question formulations/scales for practical reasons, metered data allows for potentially thousands of operationalizations. Thus, we recommend researchers using metered data to test at least several of them to assess the robustness of their results.

## Acknowledgments

The authors are very grateful to María Paula Acuña and Tracy Silva for their help with some of the tasks needed for this research.

## Author contributions

**Conceptualization:** Carlos Ochoa, Melanie Revilla.

**Data curation:** Carlos Ochoa.

**Formal analysis:** Carlos Ochoa.

**Funding acquisition:** Melanie Revilla.

**Investigation:** Carlos Ochoa, Melanie Revilla.

**Methodology:** Carlos Ochoa, Melanie Revilla.

**Project administration:** Carlos Ochoa.

**Resources:** Carlos Ochoa.

**Software:** Carlos Ochoa.

**Supervision:** Melanie Revilla.

**Validation:** Carlos Ochoa, Melanie Revilla.

**Visualization:** Carlos Ochoa.

**Writing – original draft:** Carlos Ochoa.

**Writing – review & editing:** Carlos Ochoa, Melanie Revilla.

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
