## [Decision Letter · Decision Letter 0]

14 Oct 2025

Dear Dr. Ochoa Gómez,

Thank you for submitting your manuscript to PLOS ONE. After careful consideration, we feel that it has merit but does not fully meet PLOS ONE’s publication criteria as it currently stands. Therefore, we invite you to submit a revised version of the manuscript that addresses the points raised during the review process.

We look forward to receiving your revised manuscript.

Kind regards,

Ghilan Al Madhagy Ghilan Taufiq Hail, Ph.D.

Academic Editor

PLOS ONE

**Journal Requirements: **

3. Please note that your Data Availability Statement is currently missing the repository name. If your manuscript is accepted for publication, you will be asked to provide these details on a very short timeline. We therefore suggest that you provide this information now, though we will not hold up the peer review process if you are unable.

Reviewers' comments:

Reviewer's Responses to Questions

**Comments to the Author**

1. Is the manuscript technically sound, and do the data support the conclusions?

Reviewer #1: Yes

Reviewer #2: No

Reviewer #3: Yes

2. Has the statistical analysis been performed appropriately and rigorously?

Reviewer #1: Yes

Reviewer #2: No

Reviewer #3: Yes

3. Have the authors made all data underlying the findings in their manuscript fully available?

Reviewer #1: Yes

Reviewer #2: Yes

Reviewer #3: Yes

4. Is the manuscript presented in an intelligible fashion and written in standard English?

Reviewer #1: Yes

Reviewer #2: Yes

Reviewer #3: Yes

Reviewer #1: Below we describe, point by point, the aspects we believe to be most relevant to highlight in the following investigation:

Introduction

Strengths:

• The introduction effectively contextualizes the study within the broader challenges of measuring online behavior and the limitations of survey-based approaches.

• The rationale for using metered data is well-articulated.

Suggestions for Improvement:

• The concept of “job search intensity” is introduced but not sufficiently grounded in theory. A more nuanced discussion of its dimensions (e.g., effort, persistence, engagement) would strengthen the conceptual foundation.

• Consider explicitly linking the concept to the metrics used later in the study to improve coherence between theory and operationalization.

Background and Literature Review

Strengths:

• The literature review is comprehensive and includes relevant studies on both survey-based and digital trace-based approaches to measuring job search behavior.

• The discussion of prior work on operationalization variability (e.g., Bosch, Faberman & Kudlyak) is insightful and sets the stage for the current study.

Suggestions for Improvement:

• A comparative table summarizing how job search intensity has been operationalized in previous studies could enhance clarity.

• The review could benefit from more discussion on validation strategies used in past research, especially regarding predictive or concurrent validity.

Research Question and Contribution

Strengths:

• The research question is clearly stated and addresses a significant methodological issue.

• The contribution to the literature on digital trace data and multiverse analysis is well-positioned.

Suggestions for Improvement:

• Consider elaborating on the practical relevance of the findings for researchers in applied fields (e.g., labor economics, public policy).

• The contribution could be more explicitly framed in terms of advancing best practices for digital behavioral measurement.

Data and Methods

Strengths:

• The data source (Netquest panel) and selection criteria are clearly described.

• The methodological rigor is commendable, with a systematic exploration of 10,080 operationalizations.

Suggestions for Improvement:

• Some operationalization decisions (e.g., thresholds for session separation or minimum time) appear arbitrary. Providing stronger theoretical or empirical justification would enhance credibility.

• The section is dense and technical; a summary table of operationalization factors and their levels is helpful but could be complemented with a flowchart or decision tree to aid comprehension.

Results

Strengths:

• The results are presented in a structured and detailed manner, with appropriate use of statistical analyses and visualizations.

• The analysis of convergent validity and variability across metrics is thorough.

Suggestions for Improvement:

• The interpretation of statistical findings could be made more accessible to a broader audience. For example, what does a correlation of 0.14 vs. 0.91 imply in practical terms?

• Highlighting which metrics are most robust and why would be valuable for guiding future research.

Discussion

Strengths:

• The discussion synthesizes the findings effectively and reflects on their methodological implications.

• The recognition of variability as a challenge in digital trace research is important and timely.

Suggestions for Improvement:

• The discussion could benefit from more concrete examples of how these findings might affect substantive conclusions in applied studies (e.g., gender differences in job search behavior).

• Consider offering more explicit recommendations for researchers on how to select and justify operationalizations.

Limitations

Strengths:

• The limitations are acknowledged transparently, including the reliance on a single panel and metering technology.

Suggestions for Improvement:

• The implications of these limitations for generalizability could be discussed in more depth.

• Suggesting specific avenues for future research (e.g., cross-country comparisons, integration with survey data) would strengthen this section.

Practical Implications

Strengths:

• The manuscript recognizes that operationalization decisions can influence substantive findings.

Suggestions for Improvement:

• This section is relatively brief and could be expanded. What should researchers do when faced with multiple plausible operationalizations?

• Consider including a set of “best practice” guidelines or a checklist for operationalizing behavioral constructs using metered data.

Reviewer #2: Overall Assessment

The research question is interesting and timely. However, the execution of the study does not deliver sufficient analytical depth or theoretical contribution to warrant publication. The paper remains largely descriptive and does not demonstrate the broader implications or insights that could be gained from the analysis.

Major Comments

1. The results section primarily presents correlations and variability across operationalizations, but these are descriptive statistics without deeper interpretation. There is no clear advancement in theory, methodology, or substantive knowledge about job search behavior. Readers are left wondering: what do we actually learn about job search intensity beyond the fact that different metrics yield different numbers?

2. While the paper shows variability, it does not translate this into meaningful implications for social science research or labor economics. For example, how should researchers decide among competing operationalizations? What guidance does this study provide for future measurement strategies? The discussion stops short of offering actionable recommendations.

3. Much of the paper replicates or extends Bosch (2020) and Bosch & Revilla (2019) in another context. The novelty is limited to applying the same framework to job search data. Without a stronger theoretical rationale or a substantive insight about labor market dynamics, the contribution feels incremental.

4. The paper repeatedly acknowledges that it cannot assess which operationalization is “valid.” Yet without any form of external validation (e.g., predictive validity against employment outcomes or survey responses), the exercise becomes self-contained and somewhat circular. This limits the value of the findings.

Reviewer #3: Introduction

• “The research problem is stated clearly, but it is not well justified that research is needed.” Please elaborate more and provide some arguments and support with references.

• “Language problems make it difficult to follow at times.”

• In Part 2.2, please support with more references.

Literature Review

• The current research lacks a more elaborate literature review section and supported references.

• Please elaborate more on the literature review about the operationalization factors of “job search intensity” based on current and previous research, and add the arguments to support your research if applicable.

Research Methodology

• In the Research Methodology section, the authors should identify the research type (qualitative or quantitative), the sampling procedures, and why they used each sampling method. Additionally, this section should briefly explain and reference the analytical tools.

Analysis and Discussion

• You need to state the significance of the R model analysis used in the research.

• Findings just report the results; there is no critical discussion. The analysis needs to be more deeply backed by research evidence and focus on the impact of the results.

Conclusion

• The discussion section should be elaborated on more and supported by related references (it was too short, and there was no synthesis explanation).

• Implications are included, but future directions are underdeveloped in this section.

**Do you want your identity to be public for this peer review?** For information about this choice, including consent withdrawal, please see our Privacy Policy

Reviewer #1: No

Reviewer #2: No

Reviewer #3: No

---

## [Author Response · Author response to Decision Letter 1]

7 Nov 2025

Please find our responses to the editor and reviewers in the attached document titled “Rebuttal Letter v4.docx.”

---

## [Editor Report · Decision Letter 1]

30 Nov 2025

Variability of a job search indicator induced by operationalization decisions when using digital traces from a meter

PONE-D-25-47969R1

Dear Dr. 

<table border="0" cellpadding="0" cellspacing="0" class="datatable3" style="border-collapse: collapse; width: 678.333px; line-height: 14px; background-color: rgb(255, 255, 255); color: rgb(0, 0, 51); font-family: verdana, geneva, arial, helvetica, sans-serif; font-size: 11.2px;"> <tbody> <tr style="background-color: rgb(244, 244, 244);"> <td style="padding: 3px; border: 1px solid rgb(255, 255, 255);">Carlos Ochoa Gómez,</td> </tr> </tbody></table>

We’re pleased to inform you that your manuscript has been judged scientifically suitable for publication and will be formally accepted for publication once it meets all outstanding technical requirements.

Kind regards,

Ghilan Al Madhagy Ghilan Taufiq Hail, Ph.D.

Academic Editor

PLOS ONE
---

## [Editor Report · Acceptance letter]

PONE-D-25-47969R1

PLOS One

Dear Dr. Ochoa Gómez,

I'm pleased to inform you that your manuscript has been deemed suitable for publication in PLOS One. Congratulations! Your manuscript is now being handed over to our production team.

Kind regards,

on behalf of

Dr. Ghilan Al Madhagy Ghilan Taufiq Hail

Academic Editor

PLOS One